# New Composite Sorbent for Removal of Sulfate Ions from Simulated and Real Groundwater in the Batch and Continuous Tests

**DOI:** 10.3390/molecules26144356

**Published:** 2021-07-19

**Authors:** Waqed Hassan, Ayad Faisal, Enas Abed, Nadhir Al-Ansari, Bahaa Saleh

**Affiliations:** 1Department of Civil Engineering, College of Engineering, University of Kerbala, Kerbala 56001, Iraq; waaqidh@uokerbala.edu.iq (W.H.); eng.enas.hashim@gmail.com (E.A.); 2Environmental Engineering, College of Engineering, University of Baghdad, Baghdad 10001, Iraq; 3Department of Civil, Environmental and Natural Resources Engineering, Lulea University of Technology, 97187 Lulea, Sweden; 4Mechanical Engineering Department, College of Engineering, Taif University, P.O. Box 11099, Taif 21944, Saudi Arabia; b.saleh@tu.edu.sa

**Keywords:** scrap iron, sulfate ions, activated carbon, transport, permeable reactive barrier

## Abstract

The evaluation of groundwater quality in the Dammam formation, Faddak farm, Karbala Governorate, Iraq proved that the sulfate (SO_4_^2−^) concentrations have high values; so, this water is not suitable for livestock, poultry and irrigation purposes. For reclamation of this water, manufacturing of new sorbent for permeable reactive barrier was required through precipitation of Mg and Fe hydroxides nanoparticles on the activated carbon (AC) surface with best Mg/Fe molar ratio of 7.5/2.5. Mixture of 50% coated AC and 50% scrap iron was applied to eliminate SO_4_^2−^ from contaminated water with efficiency of 59% and maximum capacity of adsorption equals to 9.5 mg/g for a time period of 1 h, sorbent dosage 40 g/L, and initial pH = 5 at 50 mg/L initial SO_4_^2−^ concentration and 200 rpm shaking speed. Characterization analyses certified that the plantation of Mg and Fe nanoparticles onto AC was achieved. Continuous tests showed that the longevity of composite sorbent is increased with thicker bed and lower influent concentration and flow rate. Computer solution (COMSOL) software was well simulated for continuous measurements. The reclamation of real contaminated groundwater was achieved in column set-up with efficiency of 70% when flow rate was 5 mL/min, bed depth was 50 cm and inlet SO_4_^2−^ concentration was 2301 mg/L.

## 1. Introduction

To maintain the quality of surface water and, consequently, the ecosystems, groundwater free of contamination is the target for several authorities and agencies that deal with water issues. Moreover, groundwater forms, in many countries, the major resource of potable water that is utilized by three out of four Europeans; Additionally, this water can be used for irrigation and industry. Spillage from gasoline storage tanks, landfills, leakage of septic tanks and accidental spills represent the point sources, while the distributed sources can be represented by infiltration of fertilizers, pesticides from agricultural regions and, in addition, contaminants in rain and snow [1].

Mostly, the detection of contamination in the groundwater needs a long time because the flow of this water is very slow and classified as a “laminar” regime. Thus, the implementation of remediation processes may be difficult and even not possible for certain locations, because the cost may exceed millions of dollars. Removing (or at least controlling) the source of contamination represents a significant step that must be applied before the remediation process. The cleanup can be achieved by one of the following methods [2,3]: containment of contaminants, pump and treat, stabilization/solidification, air stripping and natural attenuation. In addition, the permeable reactive barrier (PRB) method was presented as an alternative for treatment of polluted subsurface water and results of many studies proved that this method is effective and efficient in the reclamation of groundwater. Since the ending of the first decade from the present century, integration of the sustainability concepts and practices of green technology has been the aim of PRB studies, because it is considered by regulatory agencies as a key design element in almost every treatment strategy [4,5,6,7,8,9].

Sulfate forms the first or second most abundant anion in natural water resources [10]. Sulfur geo-chemistry represents the prime control for redox reactions of natural systems. Reduction of SO_4_^2−^ can occur extensively in the groundwater of subsurface systems. In such reduction, substantial quantity of H^+^ can be consumed, accompanied with HS^−^ production at ambient pH [11]. Popular techniques for SO_4_^2−^ elimination include: (i) Using sulfate-reducing bacteria in biological treatment, (ii) sorption processes and (iii) chemical precipitation [12]. High concentrations of SO_4_^2−^ ions result in severe problems for the environment, especially in the sectors of metallurgical, mining, chemical and agriculture [13]. Common examples of these problems include an alteration of water taste, troubles in digestion for humans and animals, acidification of soil and metals corrosion. Ions of SO_4_^2−^ are the prime contributor to water ‘‘mineralization”, thereby increasing the conductivity and corrosion of the bodies of receptors [14]. 

Layered double hydroxides (LDHs), known as hydrotalcites, have a potential role in the treatment of a wide range of anionic, organic and inorganic chemicals [15,16,17]. LDHs are inorganic substances composed of stacked layers of positively charged metal cations with charge-balancing anions situated in the interlayer sites [18]. LDH is a layered anionic that can be connected via a non-covalent bond between non-framework inter-layer anions and a positive main lamellar [19,20,21]. The surface precipitation with metal ions can be enhanced in the LDH due to availability of OH groups and the interlayer anions [22]. LDHs are a category of isostructural (2D) lamellar materials composed of brucite-like octahedral sheets with positive charges [23,24,25,26]. Magnesium/ferric-based LDH (MF-LDH) can be co-precipitated from Mg and Fe salts and are indeed very efficient and environmentally friendly sorbents/ion exchangers for anions in aqueous media [27,28,29,30]. The LDH can be used extensively in the treatment of wastewater contaminated with different chemicals [31]. Mg(OH)_2_ is an environmentally friendly solid material that recently gained great attention [32,33,34]. Fe oxide/hydroxide is also an environmentally benign solid material that has been investigated for removing phosphate from water [35,36]. Additionally, previous studies [37,38] demonstrated the long-term and environmentally safe application of iron-based oxyhydroxide adsorbents, including the competition between SO_4_^2−^, phosphate and other anions in the system. MF-LDH has a low risk of Fe(III) release into water; thus, the application of MF-LDH into water bodies is expected to be safe [39].

The usage of AC is frequently limited in the treatment of contaminated groundwater due to its high cost. Accordingly, the configuration of funnel and gate proposed by Suthersan (1999) [40] was applied for shallow groundwater using replaceable cassettes of carbon. Mixing of AC with other materials like zero-valent iron (ZVI) was an attempt that achieved to reduce the required amount (cost) of AC for treatment process [41]. In addition, the application of ZVI in remediation of groundwater was investigated at the University of Waterloo, Canada and then it had commercialized in 1992. The destructive nature through treatment processing represents the principal advantage of ZVI. This material depends on the destruction of pollutants through redox reaction, which means that the chemicals are destroyed in a redox reaction. Thus, the end-products are contained nontoxic compounds like de-chlorinated hydrocarbons, hydrogen gas and chloride in solution. This material is placed in reactive zone of a barrier; it readily oxidizes to create further electrons and surface coverage by oxygen radicals. The cations are then neutralized or driven to attach to these radicals that may be distributed over the sorbent surface in such a process called reduction. Additionally, the efficient removal of toxic metal ions from aqueous solutions was tested by using nanocomposite materials [42,43]. 

The groundwater is classified as sulfate water in the southern desert, southern region of Al-Jazira, and the strip near to the Iranian border in eastern Iraq due to the presence of gypsum rocks in the vertical and horizontal pathways of water; so, this water is not suitable for irrigation of all crops [44]. Accordingly, this study represents a good attempt for investigating the possibility of using PRB for reducing of SO_4_^2−^ ions concentration in the simulated and real groundwater to satisfy the requirements of irrigation and/or domestic uses. 

The specific aims of this work are: (1) Providing an overview of groundwater quality in the Dammam formation, Faddak farm, Karbala desert, Iraq; (2) investigation of the possibility of manufacturing a new composite sorbent consisting of scrap iron and AC coated with magnesium/iron-LDH nanoparticles for remediation of simulated and real groundwater containing SO_4_^2−^ ions; (3) identification of mechanisms governing the sorption of SO_4_^2−^ ions onto a synthesized sorbent; (4) experimental monitoring of the spatial and temporal propagation of SO_4_^2−^ along the sorbent bed and finding the capability of COMSOL software in the simulation of the measurements.

## 2. Study Area

To simulate the real properties of groundwater, several samples of this water were collected from a certain site located somewhere 19 km south-western of Karbala Governorate center under the south coast of Al-Razzaza lake, Iraq. The site is located between latitudes (43°52′44″ N), and longitudes (32°43′23″ E) with gross area reaching to 2000 dunam. The main soil type of the aquifer in this area is sediments of gravel, sand and gravelly sand with presence of clayey lenses. The area has smooth topographic features with surface elevation ranging from 20 to 61 m above sea level, decreasing from north-east to south-west, with the presence of ten wells utilized to estimate the aquifer properties. The stratigraphic column consists of Tayarat, Umm Er Radhuma, Dammam, Euphrates, Nfayil, Fat’ha, Injana, Dibdibba and Quaternary Deposits arranged from oldest to newest. More information of the composition and characteristics of these formations is presented in previous literatures like [45].

## 3. Models for Simulation of Experimental Measurements

### 3.1. Equilibrium Isotherm Models 

The sorption model is the expression correlated between the amount of contaminant loaded onto sorbent (*q_e_*, mg/g) and the concentration (*C_e_*, mg/L) remaining after the sorption process at a specific temperature and pH [46]. The utilized sorption models include [47]:Freundlich model; Equation (1) can apply for multilayer sorption onto non-homogenous surfaces as follows:
(1)qe=KFCe1n
where *K_F_* is the constant of Freundlich and 1/*n* (<1) is the intensity of sorption.Langmuir model; Equation (2) applies for homogenous surfaces and monolayer sorption:
(2)$$q_{e}=\frac{q_{max}bC_{e}}{1+bC_{e}}$$ where *q_max_* is the maximum capacity of adsorption (mg/g) and *b* is the chemical intensity on the solid matrix.

### 3.2. Kinetic Models

The transfer rate of the dissolved contaminant from aqueous to solid phases is an essential parameter to design the sorption process [48]. This rate can be calculated from the following relationships:Pseudo first order:
(3)$$q_{t}=q_{e}\left(1-e^{-k_{1}t}\right)$$
where *q_t_* and *q_e_* (mg/g) are the quantities of contaminant loaded on the solid particles at *t* and equilibrium times, respectively, and *k*_1_ is the rate constant (1/min).Pseudo second order [49]:
(4)$$q_{t}=\frac{t}{\left(\frac{1}{k_{2}q_{e}^{2}}+\frac{t}{q_{e}}\right)}$$ where *k*_2_ is the rate constant of this model (g/mg min).

### 3.3. Transport of Solute 

The dissolved contaminant transport in the subsurface environment under the effects of advection, hydro-dynamic dispersion and sorption is expressed by the “advection-dispersion-reaction” equation. The derivation of this equation was achieved by combination of Fick’s second law with the mass conservation equation. For uniform and steady velocities, the solute transport equation will be as follows for one-dimensional solute transport:(5)$$D_{z}\frac{\partial^{2}C}{\partial z^{2}}-V_{z}\frac{\partial C}{\partial z}=R\frac{\partial C}{\partial t}$$
where R is the retardation factor that is calculated with aid of the isotherm model.

## 4. Results and Discussion 

### 4.1. Evaluation of Groundwater Quality

Insightful vision about the quality of groundwater for any selected region can be conducted through the measurements of common water quality parameters like pH, EC, TDS, COD, anions and cations. However, as the target chemical adopted in this work was SO_4_^2−^ concentrations, the average spatial distribution of these ions for water samples collected from wells in the Faddak farm was changed nearly from 400 to 2100 mg/L. It is clear that the SO_4_^2−^ ions had values that exceeded the allowable limit (=250 mg/L); so, the PRB with novel reactive material can be applied to restore this water for the mentioned limit. The manufactured material consisted of AC-(Mg/Fe)-LDH mixed with scrap iron. 

### 4.2. Preparation of Novel Sorbent 

Primary tests proved that the elimination of SO_4_^2−^ from water onto AC did not exceed 1%; so, modification of AC by coating its surface with (Mg/Fe)-LDH is required. The preparation process requires addition of 20 g AC to 200 mL of aqueous solution containing Mg(NO_3_)_2_·6H_2_O and FeCl_3_·6H_2_O with ratios of 10/0, 7.5/2.5, 5/5, 2.5/7.5 and 0/10, respectively. The performance of prepared AC-(Mg/Fe)-LDH was evaluated through finding the removal of SO_4_^2−^ ions by applying 2 g sorbent to 50 mL of contaminated water with *C_o_* of 50 mg/L, speed of 200 rpm, initial pH of 7, and time of 3 h. Figure 1a explains the outputs of sorption tests where there is a dramatic variation in the removal efficiencies as a function of Mg(NO_3_)_2_·6H_2_O and FeCl_3_·6H_2_O ratio. This figure signifies that the best ratio is 7.5/2.5, which lead to maximum removal efficiency (≈35.5%); however, increasing or decreasing of the ratio from this value will cause an obvious decrease in the efficiency and the lowest value (≈1.5%) appeared at the ratio of 0/10. To improve the elimination of SO_4_^2−^ from water, the scrap iron was mixed with AC-(Mg/Fe)-LDH. A group of batch tests were implemented at the same operational conditions mentioned previously to specify the best weight ratio of coated AC to scrap iron, and results are illustrated in Figure 1b. It seems that the variation of ratio from 10/0 to 5/5 will be accompanied with an increase of efficiency from 35.5% to 59.0%, respectively; however, clear reduction in the efficiency can be observed beyond the ratio of 5/5 to be 37.5% at 0/10. All next batch and continuous tests must be conducted with reactive material consisting of mixing AC-(Mg/Fe)-LDH and scrap iron with a proportion of 5/5.

### 4.3. Effect of Operation Variables in Batch Experiments 

#### 4.3.1. Time and Initial Concentration 

The determination of equilibrium time is required for the batch study; so, the sorption onto adsorbent must be monitored with contact time. For initial pH 7 and speed of 200 rpm, sorbent (mixture of AC-(Mg/Fe)-LDH and scrap iron) dosage of 2 g was mixed with 50 mL of aqueous solution contaminated with an initial SO_4_^2−^ concentration of 50, 100, 150 and 200 mg/L, at room temperature for a contact time not greater than 3 h. For any value of *C_o_*, Figure 2a signifies that the sorption rate of SO_4_^2−^ can increase rapidly in initial times and is slowed beyond 1 h because of the reduction of binding sites. The same figure illustrates that the removal efficiencies of SO_4_^2−^ were decreased drastically due to the increase of *C_o_*. This variation may have resulted from the saturation of vacant sites interacting with SO_4_^2−^ It seems that the removal of SO_4_^2−^ was 50.32% at the equilibrium time of 1 h for *C_o_* of 50 mg/L; however, this efficiency will decrease to be 10.78% for *C_o_* of 200 mg/L for the same time. At equilibrium, results signified that there is a rapid increment in the quantity of sorbed contaminant onto sorbent in the low ranges of *C_o_*. This quantity is growing in a gradual scheme due to the increase of *C_o_*, causing an improvement in drive force for the transference of ions from liquid to solid phases; thus, a decline in the removal rate can be observed with the rise of *C_o_*.

#### 4.3.2. Initial pH

The influence of pH in the removal of SO_4_^2−^ onto prepared sorbent was examined in the range from 2.0 to 12.0 at a speed of 200 rpm, *C_o_* of 50 mg/L, dosage of 2 g/50 mL and time of 1 h (Figure 2b). The maximum adsorption of SO_4_^2−^ ions (=56%) occurred at pH 5. For pH < 5, the high electrostatic force of repulsion shows between the negative charged surface of sorbent and the negatively charged ions of SO_4_^2−^, and this caused a clear decrease in the SO_4_^2−^ uptake. This means that the increase of negatively charged sites in the adsorbent is not favorable for the uptake of SO_4_^2−^ due to the generation of repulsion forces. At higher pH, hydroxyl ions are effectively increased and, consequently, a remarkable decrease in removal SO_4_^2−^ ions can be observed. The pH change results in the dissociation of working groups of the sorbent, which lead to a shift in the equilibrium sorption properties. This can be explained in detail by measuring the point zero charge (pH_pzc_) of sorbent. Theoretically, pH_pzc_ is generally described as the pH value at which the net charge at a sorbent surface equals to zero (i.e., total positive charges = total negative charges). This phenomenon will be useful in hypothesizing the ionization mode of functional groups and the pattern of their interaction with other chemical substances in solution. In solutions with pH values lower than pH_pzc_, surfaces of solid particles are negatively charged (Figure 2c) and; therefore, they have the ability to interact with positively charged ions. While at pH values higher than pH_pzc_, the process description is the other way round. The “solid addition method” needed to be applied to find the zeta potential for the present sorbent, where pH_pzc_ is approximately equal to 4.5, as is obvious in Figure 2c. Accordingly, remaining tests were operated at pH 5 (>pH_pzc_) to ensure achievement of the maximum removal efficiency. At this pH level, being >pH_pzc_, the sorbent surface has a positive charge, which forms an electrostatic attraction with negative ions. When the pH of water is smaller than pH_pzc_, the sorbent surface has negative charges; so, it repels the negative ions. Thus, the maximum removal of SO_4_^2−^ ions occurred at an acidic pH of 5, and this result is in line with findings of previous literatures in which the maximum uptake for these ions occurred in an acidic environment with pH ranging from 3 to 6 [50,51,52].

#### 4.3.3. Sorbent Mass

The sorbent mass was changed from 0.5 to 5 g/50 mL and the removal efficiencies of SO_4_^2−^ ions were determined through the measurement of equilibrium concentration. The effect of sorbent dosage was studied under *C_o_*, pH, time and shaking speed of 50 mg/L, 5, 1 h and 200 rpm, respectively. The measurements are plotted in Figure 2d, where the removal percentages of SO_4_^2−^ ions are increased significantly with the increase of sorbent dosage. This increase is a logical trend, because the presence of a high amount of sorbent leads to more vacant sites for the interaction and capture of ions [53]. Results confirmed that the usage of 2 g of prepared sorbent can achieve the maximum removal of SO_4_^2−^ ions. It is important to calculate the mass of SO_4_^2−^ desorbed from exhausted sorbent; so, the desorption test was applied using washing reagent (represented by deionized water). Results demonstrated that the SO_4_^2−^ concentration in the washing solution was less than 2.3 mg/L at specified periods, with a total duration of 48 h. This means that the present target contaminant is incorporated with the prepared sorbent by strong forces; consequently, this sorbent has an effective property in the retention of SO_4_^2−^ ions.

### 4.4. Sorption Isotherm and Kinetics Models 

To find the suitable design for sorption system, the isotherm models for equilibrium sorption data must be determined. A set of equilibrium sorption tests for interaction of sulfate-water with AC-(Mg/Fe)-LDH mixed with scrap iron was conducted at pH 5, *C_o_* of 50 mg/L, sorbent dosage in the range of 0.5–5 g for each 50 mL, and speed of 200 rpm for time of 1 h. The outputs of these tests are simulated by models of Freundlich and Langmuir using nonlinear fitting in Microsoft Excel 2016 by applying “Solver” formula. The parameters of these models with statistical measures resulted from the fitting process were inserted in Table 1. It seems that the (*q_max_*) was 9.5 mg/g and (*b*) was 0.0028 L/mg; however, the (*K_f_*) and (1/*n*) had values of 0.0319 (mg/g) (L/mg)^1/*n*^ and 0.8963, respectively. Figure 3a shows the concurrence between the predicted curves of isotherm models and measured data. Each value of *q_e_* in this figure represents the average of three readings; however, the standard deviation for these readings must be calculated and plotted in the same figure as “error bars”. It is clear that the two isotherms are well described by the sorption data, namely the “coefficient of determination, R^2^” had the highest value and the “sum of squared errors, SSE” had the lowest value [54,55]. The (1/*n*) is smaller than unity; so, there is favorable adsorption for SO_4_^2−^ ions onto the prepared sorbent. It seems that the *q_max_* (=9.5 mg/g) is comparable with those reported for AC derived from coconut coir pith (4.9 mg/g) [56] and from Lotus leaf (9.3 mg/g) [52].

The results of kinetic tests conducted to find the time effect on removal efficiency of SO_4_^2−^ ions from water were simulated by kinetic models for *C_o_* of 50 mg/L. The constants of these models were calculated through fitting with experimental measurements using non-linear relationship—“Solver” application—in Microsoft Excel 2016. The statistical measures for evaluating the concurrence between the predicted and measured data with kinetic constants are inserted in Table 1. This table certifies that the second model is more dependable for describing ions sorption from the aqueous solution, because R^2^ > 0.97 and SSE = 0.0058. Additionally, the (*q_e_*) of SO_4_^2−^ is approached from experimental quantity and this certified the applicability of the mentioned model. Accordingly, the chemisorption must be the dominant mechanism. However, the degree of matching between the measurements and predictions of models can be observed in Figure 3b.

### 4.5. Characterization of Composite Sorbent 

The XRD analysis (Figure 4a) illustrates the composition of immobilized matrix (i.e., AC) before and after precipitation of (Mg/Fe)-LDH nanoparticles. According to standards of Joint Committee on Powder Diffraction, this figure elucidates the existence of diffraction reflection at (2θ = 23°) for AC corresponding to carbon; however, graphite appeared at reflections (2θ = 24, 26, 44, and 65°) for AC-(Mg/Fe)-LDH. In addition, the peaks (2θ = 11°) for AC-(Mg/Fe)-LDH confirm the existence of hydrotalcite-like compounds which are composed of Mg and Fe [57]. The mentioned reflections represent the new sites generated on the AC surface, which increased the affinity of the coated material towards SO_4_^2−^ ions.

Infrared spectroscopy identified the functional groups and chemical structure before and after interaction with SO_4_^2−^ ions, whereby these groups are illustrated in Figure 4b. The available spectrum is very identical to that of lignocellulosic, like rockrose and pistachio-nut shell [58]. The C–H vibrations in methylene and methyl groups can be corresponded to the band at 2929 cm^−1^. The band at 1662 cm^−1^ is attributed to groups of carbonyl C O. The vibrations of olefinic (C C) lead to the emergence of the band at 1651 cm^−1^, while two bands at 1511 and 1429 cm^−1^ can generate due to the skeletal C C vibrations in aromatic rings. The vibrations at 1463 and 1376 cm^−1^ are assigned to the bands –CH_3_ and –CH_2_– [59]. In carboxylate groups, the band at 1323 cm^−1^ can attribute to C–O vibrations. The band at 1246 cm^−1^ may attribute to esters, ethers or phenol groups. The relatively intense band at 1043 cm^−1^ can be assigned to alcohol groups (R–OH). The C–H out-of-plane bending in benzene derivative vibrations causes the band at 891 cm^−1^. The changes in the FT-IR spectra for the sorbent sample after the sorption process can be observed in Figure 4b. The sample exhibits the OH stretching vibrations band (3600–3100 cm^−1^) of surface hydroxylic groups and chemisorbed water. The asymmetry of this band at lower wavenumbers reveals the existence of strong H bonds between sorbed molecules of water and surface functional groups. The FT-IR spectra (<2000 cm^−1^) of the carbon sample demonstrate ideal absorption of structural and surface oxygen species. The bands (i) 1730–1705 cm^−1^ and (ii) 1570–1550 cm^−1^ ranges can, respectively, attribute to the C-O moieties stretching vibrations in (i) lactonic, ester, carboxylic or anhydride groups and (ii) conjugated systems like ketoesters, diketone and keto-enol structures [60]. Since water can sorb on the AC surface and non-specific interactions (physical sorption), the bands (1500–1600 cm^−1^) can be described by overlapped OH binding vibrations. The complex nature of bands in the bands (1650–1500 cm^−1^) suggests that double bond (C-C) vibrations and aromatic ring bands overlap the OH binding vibration bands and aforesaid C-O stretching vibration bands. A broad band (1470–1380 cm^−1^) is composed of sets of overlapping absorption bands ascribable to the deformation vibration of surface OH groups and in-plane vibrations of C–H.

Energy-dispersive X-ray spectroscopy (EDS) gives elemental compositional graphs for AC and composite sorbent (scrap iron + AC-(Mg/Fe)-LDH), as illustrated in Figure 5. This figure with Table 2 elucidates that the carbon forms the predominant element in the composition of AC with a percentage of 90.72%, and this is consistent with outputs of XRD analysis. Additionally, other elements, specifically O, Al, Si, S and Fe, with certain percentages can be observed in the AC composition. The table proves that the virgin AC is free of Mg, while the percentage of Fe reached to 0.62%. It seems that the percentage of Fe is increased drastically in the composite sorbent, to be 9.49% in comparison with 1.03% of Mg, and this due to the use of Fe in the coating of AC and the addition of scrap iron. Results documented that the percentage of S was increased to become 1.34% in the composite sorbent after elimination of SO_4_^2−^ from water and this certifies the occurrence of adsorption.

For 500 nm magnification scale, SEM images (Figure 6) are important for identification of the morphological properties for adopted matters after and before contact with SO_4_^2−^ ions. It can be seen that the shape of the virgin AC is irregular and nanoparticles of Mg/Fe can be recognized on the composite sorbent. The attachment of these nanoparticles on the AC surfaces made it coarse, sinuous and rougher, which is in line with surface areas determined from BET test. The surface area of AC is equal to 54 m^2^/g and this value can be increased to be 80.3 m^2^/g after coating. Obvious variations can be observed in the morphology of the composite sorbent beyond sorption process due to the attachment of SO_4_^2−^ ions.

### 4.6. Dispersion Coefficient 

For finding the relationship between the dispersion coefficient in the longitudinal direction (*D_L_*) and seepage velocity (*V*), tracer experiments mentioned previously were conducted in the column filled with composite sorbent (scrap iron + AC-(Mg/Fe)-LDH). The tests were achieved at 5, 10, 15 and 20 mL/min to ensure the flow regime in the porous medium was laminar with Reynolds number (Re) < 1–10. This type of flow is a popular regime in the groundwater flow, as mentioned by [61]. The experimental measurements are plotted and represented in the form of a linear relationship (Equation 6 correlated between *D_L_* and *V*; however, this equation is identical to Equation (7):(6)$$D_{L}=3.7386V+0.367,\;\;\;\;\;R^{2}=0.9984$$
(7)$$D_{L}=\alpha_{L}V_{L}+D^{*}$$
where *D** is the effective molecular diffusion coefficient (=$\tau D_{o}$), *α_L_* is longitudinal dispersivity, τ is tortuosity of the packed bed (=porosity, *n*) and *D_o_* is the coefficient of molecular diffusion. Using the analogy of Equation (6) with Equation (7), longitudinal dispersivity (*α_L_*) can be specified for composite sorbent used as PRB in the continuous type of tests.

### 4.7. Breakthrough Curves in the Column Tests 

#### 4.7.1. Inlet Concentration

The inlet SO_4_^2−^ concentration can influence on the propagation of chemical front through the barrier bed at ports P1 and P2 was measured for water discharge of 5 mL/min (Figure 7a,b). At lower inlet concentration, the curve is less pronounced because the sorption process occurs at the slow rate. Change of SO_4_^2−^ concentration from 50 to 100 or 150 mg/L can accompany a significant increase in the slope of the curve, and this leads to the saturation of the composite sorbent bed in a short period of time. Measurements illustrated that the lower gradient of concentration results in a slower transfer of contaminant towards the sorbent pores due to a reduction in the mass transfer coefficient [61], which increases the breakthrough and saturation times to 5% and 90% of the normalization concentration (*C/C_o_*), respectively. Figure 7a,b shows that the mentioned times are increased with decreasing inlet concentration; so, this can be accompanied with a significant increase in adsorbed quantities of the SO_4_^2−^ ions within the applied bed [62]. For example, the values of breakthrough time for 50, 100 and 150 mg/L at P1 are 137, 98 and 75 min, respectively, while they increased to be 461, 298 and 225 min, respectively, at P2 for water discharge of 5 mL/min.

#### 4.7.2. Flow Rate of Contaminated Water

The effect of water flow rate on the migration of SO_4_^2−^ ions along the composite sorbent bed was monitored by applying three values, specifically 5, 10 and 15 mL/min. Breakthrough curves (normalized concentration as function of elapsed time) under the effect of flow rate variation are plotted in Figure 7c,d at an influent concentration of 50 mg/L for ports P1 and P2. It is clear that the increase of flow rate can reduce the time required to reach breakthrough point and this will create steeper curves. This reduction in breakthrough time means that the contaminant will leave the packed bed before reaching the equilibrium status [63]. The higher discharge for the same cross sectional area is associated with an increase in the velocity of flow and this can decrease the attachment of SO_4_^2−^ ions onto the sorbent surface; therefore, an obvious reduction in removal percentage is observed [64]. Furthermore, the higher flow rate could potentially desorb some of sorbed solute; therefore, the SO_4_^2−^ ions increase rapidly, leading to an earlier breakthrough time. The present results are in line with those mentioned in previous literatures [62,65].

#### 4.7.3. Sorbent Quantity

Figure 7 illustrates the influence of bed height on the propagation of the contaminant front. For certain inlet concentration and flow rate, measurements proved that the change of bed height from 20 (P1) to 50 cm (P2) will delay the migration of contaminant, because the higher mass of the sorbent leads to the higher capacity of adsorption. This will be associated with an increase in the breakthrough time and effluent volume; so, breakthrough curves take the typical S-shape graph. In continuous experiments, the sorbent will be exhausted when the influent solution leaves the bed without remarkable remediation. The time lapse taken to attain this exhaustion stage is called “exhaust time”. It seems, from the mentioned figure, that the exhaust time of a bed increases with increasing bed height because there are more sites for adsorption. However, the capacities of adsorption are decreased with increasing sorbent mass, resulting in the overlap of binding sites [65].

#### 4.7.4. Treatment of Real Groundwater Sample

To evaluate the ability of the column packed with prepared sorbent to remediate real groundwater contaminated with SO_4_^2−^ ions, samples of water were withdrawn from wells on 1 April 2021. The measured characteristics of this sample, specifically pH, EC, TDS, TSS, SO_4_^2−^, Ca^2+^ and Mg^2+^, are listed in Table 3, which also explains their values after treatment process. The water sample was treated by injection through a column packed with 50 cm depth of scrap iron + AC-(Mg/Fe)-LDH, at a flowrate of 5 mL/min, for a contact time of 400 min. It seems that the concentration of SO_4_^2−^ ions measured for evaluation of water quality in this study was decreased drastically from 2301 to 709 mg/L, with removal efficiency approximately equal to 70%. This is considered a good result, especially as the influent was real contaminated groundwater. To attain a concentration of SO_4_^2−^ ions that is less than the acceptable limit (250 mg/L), the depth of reactive material must either increase or the flow rate can decrease to simulate the actual velocity of groundwater.

#### 4.7.5. Hydraulic Conductivity

Coefficients of hydraulic conductivity of the composite sorbent (i.e., scrap iron +AC-(Mg/Fe)-LDH) bed in the experimental column were determined under different inlet concentrations and flow rates using Darcy’s law. The coefficients had values that were approximately constant (≈ 3.3× 10^−2^ cm/s) during the period of each experiment. This indicates that the interconnected pores (i.e., effective porosity of bed) did not change through the operation process.

#### 4.7.6. Numerical Modelling

The solute equation was solved by numerical method using COMSOL Multiphysics 3.5a (2008) program (developed by Royal Institute of Technology in Stockholm, Sweden, 2005). The input constants required to achieve this solution can be summarized as follows: Bed depth = 50 cm, porosity = 0.45, dispersivity (α_L_) = 3.7386 cm and bulk density (*ρ_b_*) = 1.171 g/cm^3^. The zero concentration was the initial condition everywhere along the packed bed, while the concentration of 50 mg/L and advective flux at input and output bed, respectively, can be the boundary conditions. The solution was implemented for a one-dimensional unsteady state case to determine the contaminant concentrations along the sorbent bed. The solid lines in Figure 7 represent the breakthrough curves predicted from the COMSOL program in comparison with measurements that appeared in the symbols shape. Acceptable matching can recognize between measurements and COMSOL predictions with R^2^ exceeding 0.98 for the operational condition under consideration.

The verified model represents an efficient tool to predict the migration of SO_4_^2−^ ions under different operational conditions, in order to find the longevity of PRB able to attain the concentrations of contaminant with acceptable limits. In this regard, the magnitude of the groundwater velocity in practice is much less than that proposed in the experiments conducted; therefore, one additional contaminant interstitial velocity of 0.2 cm/min corresponding to a flow rate of 1 mL/min for the packed column under consideration was examined. Figure 8 presents a comparison between temporal distributions of the contaminant front for ports P1 and P2 due to a decrease of pore velocity (V) from 1 cm/min (flow rate (Q) = 5 mL/min) to 0.2 cm/min. It is clear that there is a delay in the migration of the contaminant front due to a decrease of velocity which causes a significant increase in the longevity of the barrier. The curve corresponding to pore water velocity of 1 cm/min is obviously more extended upward than that with 0.2 cm/min; this is due to the early emergence of the contaminant. The breakthrough times for ports P1 and P2 (Figure 8) at V = 1 cm/min were equal to 137 and 461 min, respectively; however, these times were increased drastically at V = 0.2 cm/min to be 700 and 2355 min, respectively.

## 5. Materials and Methods

### 5.1. Groundwater Quality 

The quality of groundwater is of nearly equal importance to quantity. The resource cannot be optimally used and sustained unless the quality of groundwater is assessed [66]. The quality of deep groundwater in the Faddak farm was evaluated by measuring the concentrations of SO_4_^2−^ ions, which were chosen as the target contaminant in this investigation. Water samples were collected from mentioned wells within the study area during 2019 using test tubes. Measurements of SO_4_^2−^ ions were conducted immediately beyond the taking of samples in all experiments of this work at the holy Karbala Province, Karbala Sewage directorate, Iraq using UV spectrophotometer (Shimadzu Model: UV/VIS-1650, Japan). The procedure announced in the Standard Methods for the Examination of Water and Wastewater (METHOD 9038, SULFATE (TURBIDIMETRIC)) was used to measure the concentration of SO_4_^2−^ ions [67]. According to WHO specifications, the permissible contaminant level of SO_4_^2−^ for livestock and poultry must be equal to 250 mg/L [68]. The calibration curve between the absorbance and sulfate concentration was linear with a range from 10 to 1000 mg/L. The limit of quantification (LOQ) and limit of detection (LOD) were calculated by 10 r/S and 3.3 r/S, respectively, where S is the calibration curve slope and r is the standard deviation of the regression equation (n = 10). Results proved that the LOQ and LOD had values equal to 0.035 and 0.011 mg/L respectively. 

### 5.2. Materials 

The AC was supplied from the Iraqi market with initial porosity of 0.43 (measured by evaporation method) and specific gravity of 1.211. The size distribution of particles for this material was varied from 0.6 to 1 mm. Initially, two sorption tests were applied for finding the ability of AC in the elimination of SO_4_^2−^ ions from contaminated water at a speed of 200 rpm, *C_o_* of 50 mg/L, initial pH 7, and time of 3 h for two AC dosages, specifically 1 and 2 g, added to 50 mL of water. Results proved that the percentages of removal did not exceed 1%. Hence, there is a need to increase the reactivity of this carbon through generating new binding sites. Thus, it was utilized as an immobilized solid matrix and the magnesium (Mg) with iron (Fe) hydroxides were precipitated as “nanoparticles” onto its surface to produce “AC-(Mg/Fe)-LDH”. The coefficient of hydraulic conductivity can be adopted to evaluate the suitability of the composite sorbent to be PRB beside the reactivity. 

Due to the high cost of AC [69,70], scrap iron can be tested as a low-cost sorbent to replace a part of the prepared sorbent. The usage of solid wastes in the remediation of contaminated groundwater satisfies the concept of sustainable development. The scrap is generated daily in huge quantities from industrial workshops at Baghdad city capital of Iraq. A statistical survey revealed that the eleven workshops can produce 13.2 ton of scrap per day.

To simulate the water’s SO_4_^2−^ contamination, a stock solution of 1000 mg/SO_4_^2−^ ions were prepared by dissolving Na_2_SO_4_ (obtained from HIMEDIA, India) in one liter of distilled water at room temperature, whereby its pH could be adjusted to the desirable value by addition of 0.1 M of hydrochloric acid or hydroxide of sodium. The solution with the required concentration of SO_4_^2−^ ions was obtained by “dilution process” to use it in the experimental work.

### 5.3. Synthesis of Sorbent

Sorbents with LDHs were manufactured via the co-precipitation procedure at room temperature by applying the same methodology of previous reference [16]. Groups of 200 mL aqueous solution including Mg(NO_3_)_2_·6H_2_O and FeCl_3_·6H_2_O, with various weight ratios of (Mg/Fe) (10/0, 7.5/2.5, 5/5, 2.5/7.5, 0/10), were mixed under agitation with 20 g AC to prepare AC-(Mg/Fe)-LDH. Drops of 2 M of NaOH were used to increase the pH of water to be 12, beyond agitation for 3 h, to ensure the precipitation of Mg and Fe on the surface of AC. The solid particles were isolated on the filter paper and then were washed with deionized water; thereafter, the particles were dried at 80 °C for 24 h. To prepare the mixed sorbent for elimination of SO_4_^2−^ ions, coated AC was mixed with scrap iron in the different proportion percentages as follows: 100/0, 75/25, 50/50, 25/75, 0/100. The suitability of the final sorbent was detected by measuring the removal efficiency of SO_4_^2−^ under different operational conditions. 

### 5.4. Characterization Analyses

The examination of crystalline structure for the synthesized sorbent was conducted via X-ray diffraction (XRD) analysis (Siemens X-ray diffractometer, D8 Advance, Bruker, Germany). Analysis of FT-IR specifies the functional groups that enhanced the sorption of SO_4_^2−^ ions and this test is important in the identification of predominant mechanisms. Scanning electron microscopy (SEM) with an EDS (XFlash 5010; Bruker AXS Microanalysis, Berlin, Germany) was operated to examine the surficial morphology and topography of sorbent under 21 °C and relative humidity of 55–60%. The zeta potential of manufactured sorbent was measured by Zeta Potential Analyzer (Zeta-Meter Inc., Harrisonburg, VA, USA).

### 5.5. Batch Mode Operation

Interaction of water contaminated with SO_4_^2−^ ions was firstly conducted with AC-(Mg/Fe)-LDH to specify the best weight ratio of (Mg/Fe) that must be added to ensure the maximum removal percentage of SO_4_^2−^. Thereafter, specification of the percentages of AC-(Mg/Fe)-LDH and scrap iron that must be mixed together was required, along with specification of the highest removal amount that can be adopted in order to identify the suitable values of these percentages. Results stated that the best (Mg/Fe) weight ratio was 7.5/2.5 and AC-(Mg/Fe)-LDH must be mixed with scrap iron in the proportion ratio of 50:50 to obtain the final composite (or mixed) sorbent. To optimize the operational parameters, this sorbent must be tested with various values of agitation time, initial pH, sorbent dosage and initial concentration (*C_o_*). The tests require the preparation of a set of 250 mL flasks filled with 50 mL of water containing 50 mg/L SO_4_^2−^ ions. Different masses of sorbent were added and the flasks were stirred at 200 rpm on an orbital shaker (SK-300, Biobase Biodustry (Shandong) Co., China) for 3 h. Then, the solid particles were separated from treated water using filter papers (42, whatman, England).

The residual concentration (*C_e_*) of the SO_4_^2−^ remaining in the treated water was measured by UV spectrophotometer. Tests were conducted under pH (2–12), *C_o_* (50–200 mg/L), and sorbent mass (*m*, 0.5–5 g) added to a volume (*V*, 50 mL) for duration not greater than 3 h. The following equation can applied to determine the SO_4_^2−^ sorbed onto the sorbent (*q_e_*) [71]:(8)$$q_{e}=\left(C_{o}-C_{e}\right)\frac{V}{m}$$

The isotherm can draw between determined *q_e_* and *C_e_*. Additionally, the variations of *q_e_* with contact time were established and these kinetic data can be analyzed to specify the predominant mechanism. The efficiency of removal (*R*) is calculated as follows:(9)$$R=\frac{\left(C_{o}-C_{e}\right)}{C_{o}}\times 100$$

### 5.6. Continuous Mode Operation

The experimental set-up manufactured in this work to investigate the ability of mixed sorbent (i.e., 50% AC-(Mg/Fe)-LDH + 50% scrap iron) to trap SO_4_^2−^ ions within contaminated groundwater consisted of two acrylic columns. Each column had the dimension: Length equal to 50 cm and inner diameter of 2.5 cm. Two ports, P1 and P2, were supplied for each column at 20 and 50 cm, respectively, from the base, as shown schematically in Figure 9. Column tests were used to simulate the movement of chemical in one-dimension to represent the actual situation for operating a barrier where the contaminated water flowed in an upward direction. Effluent normalized concentration of contaminant and barrier permeability were the criteria adopted for evaluating the performance of the reactive media.

The contaminated water was injected by peristaltic pump (Fischer Scientific, FB-70382, Germany) through the column with water discharge of 5, 10, or 15 mL/min that was checked using a flow meter. The propagation of SO_4_^2−^ ions front was monitored at different values of inlet concentrations (50, 100, and 150 mg/L), adsorbent bed heights (20 and 50 cm) and water flow rates. At specified periods, the samples of water were withdrawn from the mentioned ports using a needle. Additionally, the coefficient of hydraulic conductivity (K) was measured by constant head method (ASTM D2434–68) at room temperature [61]. The K is determined as follows:(10)K=QA∆h∆l
where A is the area of bed-cross section, Q is the discharge (rate) of flow and Δh is the head difference along the bed length (Δl). To measure the longitudinal dispersion coefficient (*D_L_*), the tracer test requires the usage of deionized water containing 1 g/L sodium chloride and this water must be pumped to the bed at discharge rates of 5, 10, 15 and 20 mL/min. The *D_L_* is calculated using Equation (11) [72]:(11)$$D_{L}=\frac{1}{8}{\left[\frac{\left(z_{o}-Vt_{0.16}\right)}{{\left(t_{0.16}\right)}^{0.5}}-\frac{\left(z_{o}-Vt_{0.84}\right)}{{\left(t_{0.84}\right)}^{0.5}}\right]}^{2}$$
where *V* is the pore velocity, *z_o_* is the bed depth (L), while *t*_0.84_ and *t*_0.16_ represent the periods required to reach 84% and 16% of C/C_o_, respectively.

## 6. Conclusions 

To decrease the SO_4_^2−^ ions in the groundwater to be less than the acceptable limit, PRB can be applied by manufacturing new reactive material. This material (consisting of 50% scrap iron plus 50% AC-(Mg/Fe)-LDH) was efficient in the elimination of these ions with efficiency exceeding 59.0% for the Mg/Fe ratio of 7.5/2.5. Batch tests proved that the best operational conditions for interaction of this sorbent and artificial contaminated water were initial pH 5, *C_o_* of 50 mg/L, time of 1 h, and sorbent dosage of 2 g/50 mL at 200 rpm. Langmuir and Freundlich models were able to represent the sorption isotherm measurements with R^2^ and SSE of > 0.92 and < 0.0043, respectively. The *q_max_* was 9.5 mg/g and the intensity constant was 0.0028 L/mg. Outputs of XRD, FT-IR and SEM-EDS tests certified that the Mg and Fe hydroxides were precipitated on the surface of AC. After coating, the XRD test confirmed the existence of hydrotalcite-like compounds which were composed of Mg and Fe, while EDS proved the presence of a clear increment in the percentage of these elements. The surface area of AC was equal to 54 m^2^/g and this value could be increased to be 80.3 m^2^/g after coating. Outputs of continuous tests proved that the inlet concentration of SO_4_^2−^ ions, rates of flow and depth of bed had a remarkable effect on the shape of the contaminant front. A higher depth with lower rate of flow and inlet concentration can delay the appearance of the SO_4_^2−^ front along the length of the bed and this will increase the longevity of the barrier. The prepared sorbent in the column operation mode was effective in the elimination of SO_4_^2−^ ions from real contaminated water, with removal efficiency reaching up to 70%. COMSOL can estimate the PRB efficacy due to variation of inlet concentration, bed depth and flow rate, with obvious matching between the numerical predictions and measurements (R^2^ ≥ 0.98).

## Figures and Tables

**Figure 1 molecules-26-04356-f001:**
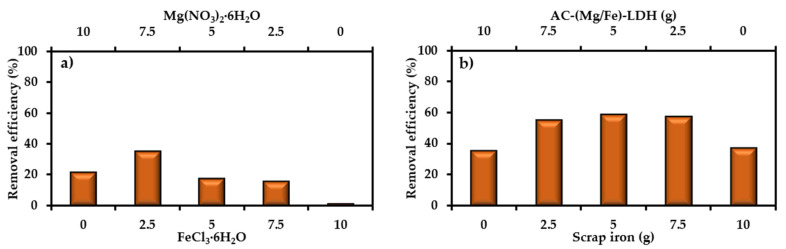
Removal efficiency of SO_4_^2−^ ions from contaminated water onto (**a**) AC-(Mg/Fe)-LDH for different weight ratios of Mg(NO_3_)_2_·6H_2_O and FeCl_3_·6H_2_O and (**b**) mixture of AC-(Mg/Fe)-LDH and scrap iron.

**Figure 2 molecules-26-04356-f002:**
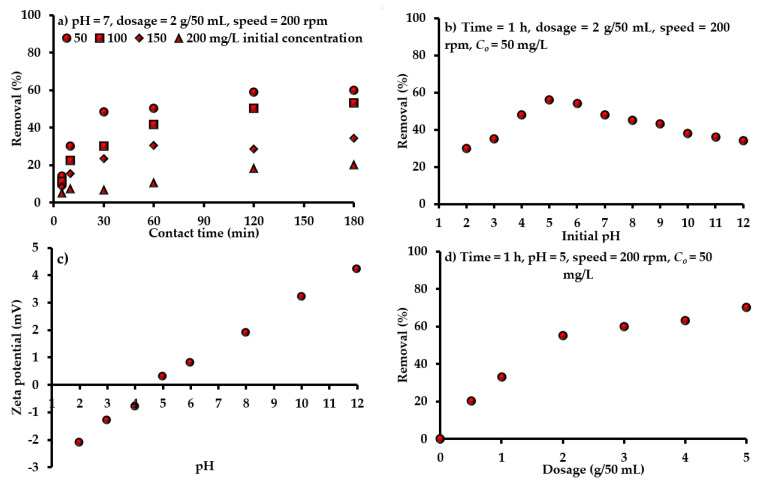
Influence of (**a**) time and *C_o_*, (**b**,**c**) pH and (**d**) sorbent dosage on the efficacy of prepared composite sorbent under different operational conditions.

**Figure 3 molecules-26-04356-f003:**
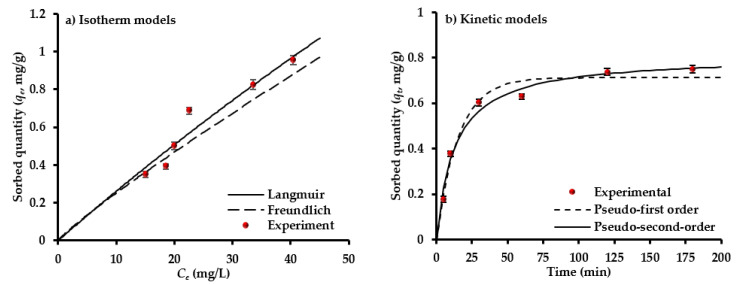
Sorption isotherms (**a**) and kinetics models (**b**) for the interaction of composite sorbent and solution containing SO_4_^2−^ ions.

**Figure 4 molecules-26-04356-f004:**
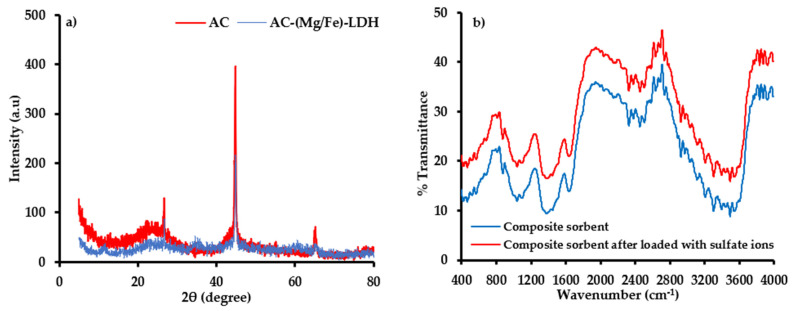
(**a**) Pattern of XRD for structure of AC and AC-(Mg/Fe)-LDH and (**b**) FT-IR test for (scrap iron +AC-(Mg/Fe)-LDH) sorbent before and after interaction with SO_4_^2−^ ions.

**Figure 5 molecules-26-04356-f005:**
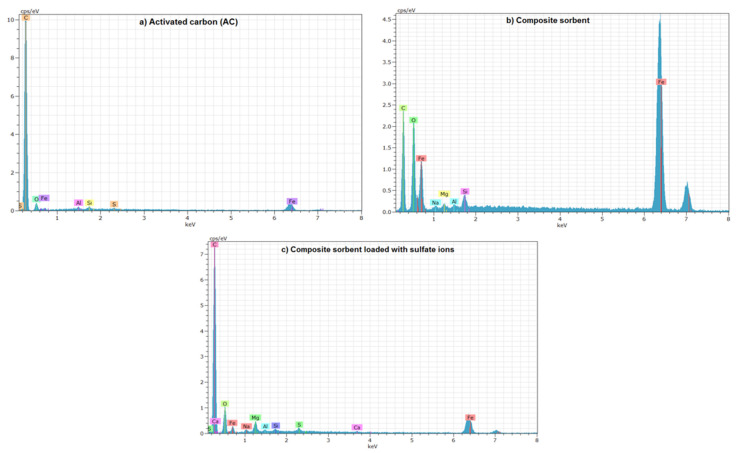
EDS spectrum of (**a**) AC, (**b**) composite sorbent before sorption and (**c**) composite sorbent after sorption of SO_4_^2−^ ions.

**Figure 6 molecules-26-04356-f006:**
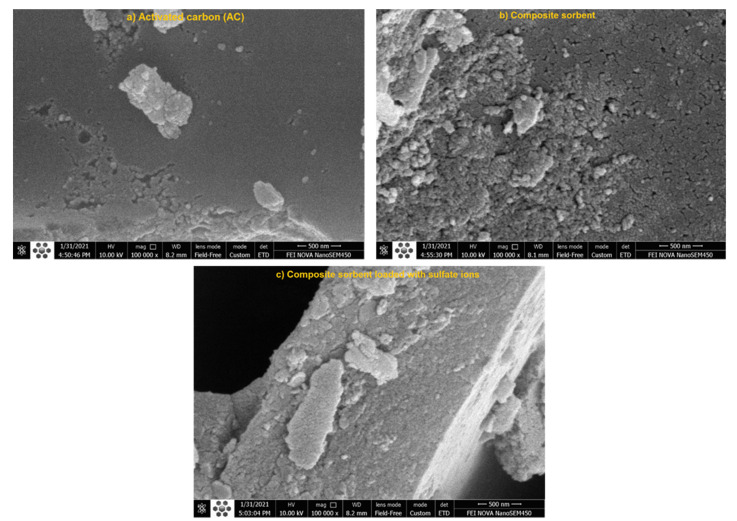
Images from SEM analysis for (**a**) AC, (**b**) composite sorbent before sorption and (**c**) composite sorbent after sorption of SO_4_^2−^ ions at 500 nm magnification scale.

**Figure 7 molecules-26-04356-f007:**
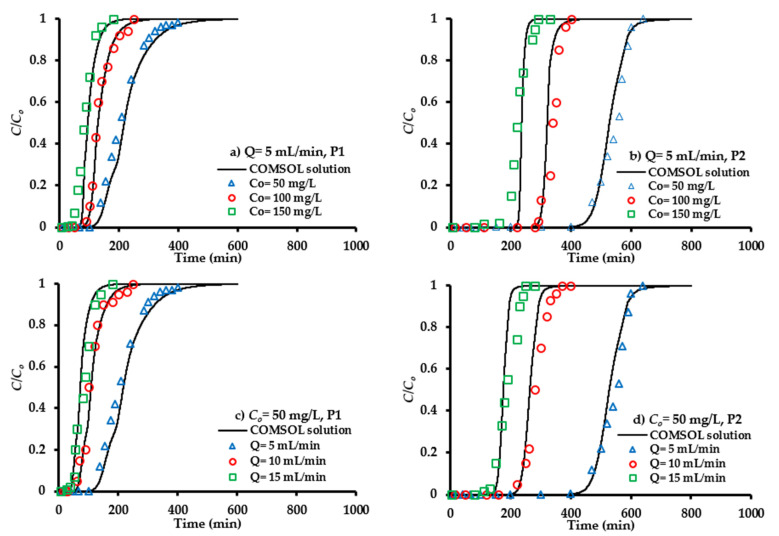
COMSOL outcomes and measurements for the SO_4_^2−^ ions normalized concentration at various magnitudes of inlet concentrations (**a**,**b**) and flow rates (**c**,**d**) for P1 and P2 ports.

**Figure 8 molecules-26-04356-f008:**
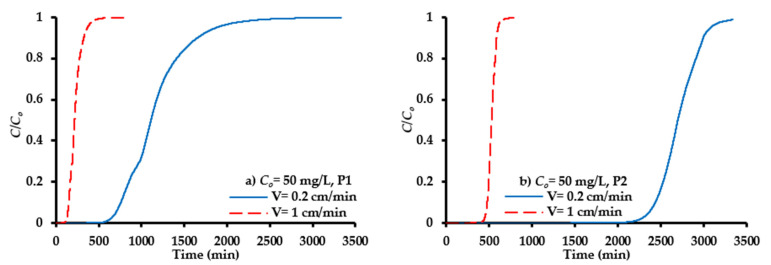
COMSOL prediction for propagation of SO_4_^2−^ ions front under variation of pore water velocity for ports (**a**) P1 and (**b**) P2.

**Figure 9 molecules-26-04356-f009:**
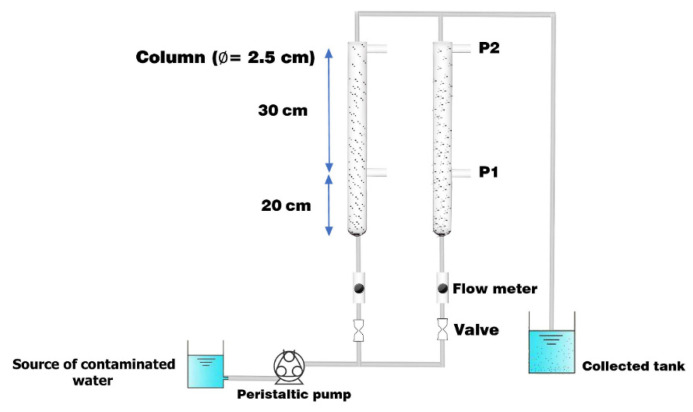
Components of laboratory set-up for continuous mode operation.

**Table 1 molecules-26-04356-t001:** Constants of equilibrium and kinetic models for removal of SO_4_^2−^ ions from contaminated water onto the prepared sorbent.

Model	Parameter	Value
Freundlich	*K_F_* (mg/g)(L/mg)^1/*n*^	0.0319
	1/*n*	0.8963
	R^2^, SSE	0.9253, 0.0430
Langmuir	*q_max_* (mg/g)	9.5
	*b* (L/mg)	0.0028
	R^2^, SSE	0.9265, 0.0222
Pseudo first-order	*q_exp._* (mg/g)	0.7700
	*k*_1_ (min^−1^)	0.0653
	*q_e_* (mg/g)	0.7224
	R^2^, SSE	0.9656, 0.0087
Pseudo second-order	*k*_2_ (g/mg min)	0.0963
	*q_e_* (mg/g)	0.8072
	R^2^, SSE	0.9782, 0.0058

**Table 2 molecules-26-04356-t002:** Percentages of elements in the AC and composite sorbent before and after sorption of SO_4_^2−^ ions specified by EDS analysis.

Element (%)	AC	Composite Sorbent	
		Before Sorption	After Sorption
C	90.72	56.14	79.25
O	8.5	32.75	18.69
Mg	0.0	1.03	0.56
Al	0.07	0.17	0.05
Si	0.06	0.42	0.05
S	0.03	0	1.34
Fe	0.62	9.49	1.05

**Table 3 molecules-26-04356-t003:** Values of popular characteristics for real groundwater samples before and after interaction with composited sorbent prepared in this work in column mode operation.

Parameter	Raw Groundwater	Treated Groundwater
pH	7	8.3
EC (µS/cm)	5770	4561
TDS (mg/L)	1220	1170
TSS (mg/L)	21	16
SO_4_^2−^ (mg/L)	2801	709
Ca^2+^ (mg/L)	370	188
Mg^2+^ (mg/L)	34	29

## Data Availability

The data presented in this study are included in the article. Further inquiries can be directed to the corresponding author.

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
