# Peer review of "New Composite Sorbent for Removal of Sulfate Ions from Simulated and Real Groundwater in the Batch and Continuous Tests"

_molecules, 2021, doi:10.3390/molecules26144356_

Round 1
Reviewer 1 Report
I have carefully read the manuscript MDPI Molecules-1254524 and find the work relevant, fitting the scope of the journal and likely to be of interest to a broad readership. The manuscript, however, needs some major revisions to be considered as publishable in the journal MDPI Molecules. In order to improve the quality of the manuscript, authors may take the following recommendations and comments into consideration:
1) English needs to be polished and grammar must be improved significantly throughout the whole manuscript, especially the first half of the manuscript.
2) Abstract, Lines 19-21: Eliminate sulfate from a synthetic sulfate solution? Maximum adsorption capacity of 9.4627 mg/g – this high precision with four digits after the comma does not make sense – better round up to 9.5 mg/g. For the rest of the variables, write the value immediately after you mention the variable otherwise it becomes unnecessarily confusing, i.e. reformulate as: “… for time 1 h, sorbent dosage 40 g/L (normalize the 2 g/50 mL to g/L) and initial pH 5 at 50 mg/L initial sulfate concentration (does this value refer to sulfate SO4 or to the elemental sulfur SO4-S concentration?) and 200 rpm shaking speed.” Same comment applies to Line 27: write the values immediately after you mention the variables to avoid confusion. Also specify if 2301 mg/L refers to SO4 or SO4-S.
3) Introduction, Line 68: “… stacked layers of positively charged metal cations …”
4) Lines 74-76: Magnesium/ferric-based LDH (MF-LDH) can be co-precipitated from Mg and Fe salts and are indeed very efficient and environmentally friendly sorbents / ion exchangers for anions in aqueous media. A very good study demonstrating not only the synthesis but also the application of MF-LDH in wastewater treatment are recommended here:
Mandel, K. et al. (2013). Layered double hydroxide ion exchangers on superparamagnetic microparticles for recovery of phosphate from waste water. Journal of Materials Chemistry A, 1 (5), 1840−1848. DOI: https://doi.org/10.1039/C2TA00571A
5) Lines 77-79: As authors stated, Fe oxide/hydroxide is also an environmentally benign solid material that has been investigated for the adsorptive removal of phosphate in water. There is extensive research on this topic, but I would highlight two exceptionally comprehensive pilot-scale studies which demonstrate the long-term and environmentally safe application of iron-based oxyhydroxide adsorbents, including the competition between sulfate, phosphate and other anions in the system:
Drenkova-Tuhtan, A. et al. (2021). Sorption of recalcitrant phosphonates in reverse osmosis concentrates and wastewater effluents – influence of metal ions. Water Science & Technology (2021), 83 (4), 934-947. DOI: https://doi.org/10.2166/wst.2021.026.
Drenkova-Tuhtan, A. et al. (2017). Pilot-scale removal and recovery of dissolved phosphate from secondary wastewater effluents with reusable ZnFeZr adsorbent @ Fe3O4/SiO2 particles with magnetic harvesting. Water Research, 109, 77−87: https://doi.org/10.1016/j.watres.2016.11.039.
6) Introduction too long and detailed. Certain paragraphs, e.g. Lines 81-101, can be cut shorter by reducing the given fundamental information not directly related to the research. Study area description can also be shortened. Less important details can be moved to a supplementary material.
7) Materials and Methods, Lines 139-141: Authors need to specify which standard they followed for the spectrophotometric measurement of sulfate ions. Also, the definition “clean polyethylene bottles” is a subjective non-scientific term. How were the bottles cleaned to assure lack of cross-contamination and how long was the groundwater stored in these PE bottles and under what conditions before the experiments were conducted?
8) Lines 149-150: Authors state that the pure activated carbon had negligible sulfate removal efficiency (ca. 1%), but they do not show any of these results. Maybe they can include them in a supplementary material.
9) Section 3.2. “Materials” is also too long and can be shortened. Authors are already discussing some results (e.g. the increased sulfate removal efficiency 35.5%) which does not belong to the Materials and Methods section, but rather to the Results and Discussion. The rest of the text in this section can also be revised only to the point and shortened. Replace “ambient of the room” with “room temperature”. As mentioned already in Point 1, English and grammar need revision and significant improvement.
10) In Section 3.3. “Synthesis of sorbent” which exact procedure did the authors follow for the LDH co-precipitation method? Quote a literature reference here.
11) Section 4.2. “Transport of solute”: All this theory can be removed from the main text or moved to the supplementary material. It is redundant and not used directly in calculations in the Results and Discussion section. Generally, all hydraulic and hydrodynamic parameters mentioned in the Methods section are not the main focus of the actual experimental research presented in this work and can be moved to supplementary. This paper focuses rather on water quality and remediation.
12) Section 5.1. “Evaluation of groundwater quality”: So, the authors did not measure any other physio-chemical parameters, besides sulfate, of the groundwater used in the study? If they did measure, it would be useful to include a full characterization of the groundwater with all measured parameters in a table format to understand better the competing reaction that may take place due to competing ions.
13) Figure 3a: Did authors perform any modeling of the sulfate sorption kinetics, e.g. pseudo-first or pseudo-second order kinetic models? If yes, please include the calculated model parameters, model constants and add kinetic fitting curves.
14) In each sub-figure in Figure 3 authors are varying one parameter at a time to see the effect on sulfate removal, but all other parameters are fixed. The fixed parameters need to be specified with their values in the respective figure caption. For instance, in Figure 3a the variable is the contact time, but the other parameters such as pH, initial sulfate concentration, adsorbent dose, etc. are fixed and should be listed with their values in the figure caption to give the reader a better overview.
15) Figure 3c: How was the Zeta potential (mV) measured instrumentally? Needs to be described in the Materials and Methods section.
16) Section 5.5. “Characterization of composite sorbent” should come logically first, prior to the previous sections 5.3. “Effect of operation variables in batch experiments” and 5.4. “Sorption isotherms”. Better switch the order of these sections for a more logical flow of the narrative.
17) Lines 564-566: Better present the measured characteristics of the real groundwater sample in a tabular format – easier for the reader to follow. In the same table combine with the effluent characteristics after treatment in the column, i.e. with the values listed in Line 569.
18) “Dramatically” is not a scientific term and it appears a few times throughout the manuscript! Better replace with “drastically” or something else.
19) Did authors test the number of bed volumes the novel composite sorbent can reach under optimal operating conditions? Was also the reusability of the sorbent tested, i.e. the ability to be regenerated and reused again?
20) What about the mechanical and chemical stability of the proposed composite sorbent material? Did authors notice any leaching effects, i.e. any of the precursor sorbent chemicals leached into the treated effluent?
Reviewer 2 Report
Manuscript Number: molecules-1254524
|
|
|
Minor drawbacks and recommended improvements |
|
1 |
Line 60, page 2 |
The authors should replace the word sulfate ions (anions) with SO42-. This abbreviature was incorporated in line 290, page 7. Please incorporate this abbreviate in all manuscripts. |
|
2 |
Line 154, page 4 |
The authors should replace the word activated carbon with AC. This abbreviature was incorporated in line 81, page 2. Please incorporate this abbreviate in all manuscripts. |
|
3 |
Line 132, page 3 Line 258, page 6 |
Please carefully check the whole article where machines/software/chemicals are first mentioned. Add the information of company, city, country. |
|
|
Line 141, page 3 Line 209, page 5 Line 231, page 5 Line 223, page 5 Line 585, page 16 |
Add the information of company, city, country. |
|
4 |
Line 237, page 5 |
The authors should indicate references. |
|
5 |
Line 637 and 641 |
The authors should move figures 8 and 9 to section 5.7.6. |
|
|
|
Major drawbacks and recommended improvements |
|
6 |
Line 142, page 3 |
In relation to the analytical methodology for quantification of sulfate ions, the authors should indicate the analytical quality parameters (limit of detection (LOD), the limit of quantification (LOQ), sensibility (S) and linearity). |
|
7 |
Line 399, page 10 |
Sorption experiments were performed with replicates?. The results in Table 1 do not show SE. |
|
|
Figure 3 a) |
The authors could calculate i) qmáx, t1/2 by mean of sorption kinetics models (pseudo –first or second order model, ii) to evaluate the homogeneity or heterogeneity of sorbent by mean of Elovich model, iii) sulfate ions sorption mechanisms by mean of IPD model; and iv) kind of sorption sites of the sorbent (Type 1 sites (where sorption is assumed to be instantaneous, F close to 1) or Type 2 sites (where sorption is considered time-dependent) by mean of Two-Site Non-Equilibrium (TSNE) model. |
Round 2
Reviewer 1 Report
The revised manuscript MDPI Molecules-1254524-v2 fulfills all expected revisions supposed to be made and is now of good publishable quality. The authors have carefully addressed every single comment and remark from the reviewers. The answers are elaborate, extensive and provide enough arguments and proof to validate the authors' statements. All necessary changes have been made based on the reviewers' recommendations. In my opinion, in this revised form the manuscript should be accepted for publication in MDPI Molecules.
Author Response
Dear Editor:
We highly appreciate the detailed valuable comments of the reviewer on our manuscript with number molecules-1254524 and entitled “New composite sorbent for removal of sulfate ions from simulated and real groundwater in the batch and continuous tests”.
Comment 1)
The revised manuscript MDPI Molecules-1254524-v2 fulfills all expected revisions supposed to be made and is now of good publishable quality. The authors have carefully addressed every single comment and remark from the reviewers. The answers are elaborate, extensive and provide enough arguments and proof to validate the authors' statements. All necessary changes have been made based on the reviewers' recommendations. In my opinion, in this revised form the manuscript should be accepted for publication in MDPI Molecules.
Response
Thank you so much for kind comments
Sincerely,
Prof. Dr. Ayad A.H. Faisal
Professor in Environmental Engineering – Corresponding author.
Iraq/University of Baghdad/College of Engineering/Environmental Engineering Department
E-mail: ayadabedalhamzafaisal@yahoo.com

Reviewer 2 Report
Manuscript Number: molecules-1254524
|
|
|
Minor drawbacks and recommended improvements |
|
|
|
|
1 |
Line 60, page 2 |
The authors should replace the word sulfate ions (anions) with SO42-. This abbreviature was incorporated in line 290, page 7. Please incorporate this abbreviate in all manuscript. |
The authors incorporated this suggestion into the revised version. |
|
|
|
2 |
Line 154, page 4 |
The authors should replace the word activated carbon with AC. This abbreviature was incorporated in line 81, page 2. Please incorporate this abbreviate in all manuscript. |
The authors incorporated this suggestion into the revised version. |
|
|
|
3 |
Line 132, page 3 Line 258, page 6 |
Please carefully check the whole article where machines/software/chemicals are first mentioned. Add the information of company, city, country. |
The authors incorporated this suggestion into the revised version. |
|
|
|
|
Line 141, page 3 Line 209, page 5 Line 231, page 5 Line 223, page 5 Line 585, page 16 |
Add the information of company, city, country. |
The authors incorporated this suggestion into the revised version. |
|
|
|
4 |
Line 237, page 5 |
The authors should indicate reference. |
The authors incorporated this suggestion into the revised version. |
|
|
|
5 |
Line 637 and 641 |
The authors should move figures 8 and 9 to section 5.7.6. |
The authors incorporated this suggestion into the revised version. |
|
|
|
|
|
Major drawbacks and recommended improvements |
|
|
|
|
6 |
Line 142, page 3 |
In relation to analytical methodology for quantification of sulfate ions, the authors should indicate the analytical quality parameters (limit of detection (LOD), the limit of quantification (LOQ), sensibility (S), and linearity). |
The authors do not incorporate the quality parameters (limit of detection (LOD), the limit of quantification (LOQ), sensibility (S), and linearity). |
|
|
|
7 |
Line 399, page 10 |
Sorption experiments were performed with replicates?. The results in Table 1 do not show SE. |
|
|
|
|
|
Figure 3 a) |
The authors could calculate i) qmáx, t1/2 by mean of sorption kinetics models (pseudo –first or second order model, ii) to evaluate the homogeneity or heterogeneity of sorbent by mean of Elovich model, iii) sulfate ions sorption mechanisms by mean of IPD model; and iv) kind of sorption sites of the sorbent (Type 1 sites (where sorption is assumed to be instantaneous, F close to 1) or Type 2 sites (where sorption is considered time-dependent) by mean of Two-Site Non-Equilibrium (TSNE) model. |
The authors incorporated the pseudo –first and pseudo-second order models. |
|
|
Author Response
Dear Editor:
We highly appreciate the detailed valuable comments of the reviewer on our manuscript with number molecules-1254524 and entitled “New composite sorbent for removal of sulfate ions from simulated and real groundwater in the batch and continuous tests”.
The reviewer’s comments are highly insightful and enabled us to greatly improve the quality of our manuscript. We have fully addressed each reviewer’s comments point by point to improve and reconstruct the quality of our paper. The manuscript has revised carefully and the revisions as per reviewer’s concern and some other essential modifications in the text are shown by Red Color. We hope that the editor and reviewer will be satisfied with our responses and look forward for the acceptance of this revised manuscript.
Comment 1)
Line 142, page 3
In relation to the analytical methodology for quantification of sulfate ions, the authors should indicate the analytical quality parameters (limit of detection (LOD), the limit of quantification (LOQ), sensibility (S) and linearity).
Response
To satisfy this comment, the following phrase is added to “3.1. Groundwater quality analysis” section as follows:
(The calibration curve between the absorbance and sulfate concentration was linear with range from 10 to 1,000 mg/L. The limit of quantification (LOQ) and limit of detection (LOD) have calculated by 10 r/S and 3.3 r/S, respectively, where S is the calibration curve slope and r is the standard deviation of the regression equation (n = 10). Results proved that the LOQ and LOD have values equal to 0.035 and 0.011 mg/L respectively.)
Comment 2)
Line 399, page10
Sorption experiments were performed with replicates? The results in Table 1 do not show SE.
Response
To satisfy this comment, the following phrase is added to “5.4. Sorption isotherm and kinetics models” section as follows:
(Each value of qe in this figure represents the average of three readings; however, the standard deviation for these readings must be calculated and plotted in the same figure as “error bars”.)
Sincerely,
Prof. Dr. Ayad A.H. Faisal
Professor in Environmental Engineering – Corresponding author.
Iraq/University of Baghdad/College of Engineering/Environmental Engineering Department
E-mail: ayadabedalhamzafaisal@yahoo.com
